


# 1 Data inversion methods to determine sub-3 nm aerosol size
# 2 distributions using the Particle Size Magnifier

Runlong Cai[1,2,#], Dongsen Yang[3,#], Lauri R. Ahonen[2], Linlin Shi[3], Frans Korhonen[2], Yan Ma[3], Jiming
Hao[1], Tuukka Petäjä[2], Jun Zheng[3,*], Juha Kangasluoma[2], and Jingkun Jiang[1,*]
[1] State Key Joint Laboratory of Environment Simulation and Pollution Control, School of Environment, Tsinghua
University, 100084 Beijing, China
[2] Institute for Atmospheric and Earth System Research / Physics Faculty of Science, University of Helsinki, P.O. Box 64,
00014 Helsinki, Finland
[3] Collaborative Innovation Center of Atmospheric Environment and Equipment Technology, Nanjing University of
Information Science & Technology, 210044 Nanjing, China
#: Runlong Cai and Dongsen Yang contribute equally to this work
*: *Correspondence to*: J. Jiang (jiangjk@tsinghua.edu.cn) and J. Zheng (zheng.jun@nuist.edu.cn)
**Abstract.** Measuring particle size distribution accurately down to approximately 1 nm is needed for studying atmospheric
new particle formation. The scanning particle size magnifier (PSM) using diethylene glycol as the working fluid has been
used for measuring sub-3 nm atmospheric aerosol. A proper inversion method is required to recover the particle size
distribution from PSM raw data. Similar to other aerosol spectrometers and classifiers, PSM inversion can be deduced to
a problem described by the Fredholm integral equation of the first kind. We tested the performance of the step-wising
method, the kernel function method (Lehtipalo et al., 2014), the H&A linear inversion method (Hagen and Alofs, 1983),
and the expectation-maximization (EM) algorithm. The step-wising method and the kernel function method were used in
previous studies on PSM. The H&A method and the expectation-maximization algorithm were used in data inversion for
the electrical mobility spectrometers and the diffusion batteries (Maher and Laird., 1985), respectively. In addition, Monte
Carlo simulation and laboratory experiments were used to test the accuracy and precision of the particle size distributions
recovered using four inversion methods. When all of the detected particles are larger than 3 nm, the step-wising method
may report false sub-3 nm particle concentrations because of assuming an infinite resolution, while the kernel function
method and the H&A method occasionally reports false sub-3 nm particles because of using the unstable least square
method. The accuracy and precision of the recovered particle size distribution using the EM algorithm are the best among
the tested four inversion methods. Compared to the kernel function method, the H&A method reduces the uncertainty
while keeping a similar computational expense. The measuring uncertainties in the present scanning mode may contribute
to the uncertainties of the recovered particle size distributions. We suggest using the EM algorithm to retrieve the particle
size distributions using the particle number concentrations recorded by the PSM. Considering the relatively high
computation expenses of the EM algorithm, the H&A method is recommended to be used for preliminary data analysis.
We also gave practical suggestions on PSM operation based on the inversion analysis.



**1 Introduction**
The particle size magnifier (PSM) using diethylene glycol as the working fluid (Vanhanen et al., 2011) is widely used in
new particle formation studies (Kulmala et al., 2012; Kulmala et al., 2013; Kontkanen et al., 2017) and other industrial
applications (Nosko et al., 2016; Ahonen et al., 2017). A PSM can report particle size distributions in the 1-3 nm size
range, which is a key size region in the nucleation study. Particles in the PSM grow into larger sizes due to the
condensation of super saturated diethylene glycol, and these particles after the initial growth are detected using a
downstream condensation particle counter (CPC). The PSM detection efficiency (the CPC is included if not specially
mentioned) of particles with a certain diameter is a function of the super saturation ratio of diethylene glycol. Increasing
the flow rate passing through the chamber containing saturated diethylene glycol vapour, i.e., the saturator flow rate, can
enhance the super saturation ratio thus the particle detection efficiencies. The total particle number concentration detected
by the PSM varies with the varying saturator flow rate, and one can determine the particle size distribution according to
the observed relationship between the particle number concentration and the saturator flow rate.
A proper inversion method is required to recover the particle size distribution using the recorded relationship between the
particle number concentration and the saturator flow rate. The step-wising method and the kernel function method were
used in previous studies for PSM inversion (Lehtipalo et al., 2014). The step-wising method is a one-to-one linear
inversion method using the relationship between the 50% cut-off size and the saturator flow rate, which essentially
assumes infinite sizing resolutions, i.e., the particles of a specific size are activated at a certain saturator flow rate.
However, such an approximation may lead to non-negligible errors due to the relatively low resolution of the PSM. The
kernel function method accounts for the detection efficiency curves, and the particle size distribution is recovered using
the non-negative least square method.
Although the uncertainties of the particle size distribution determined using the PSM was discussed recently
(Kangasluoma and Kontkanen, 2017), the uncertainties introduced during the data inversion have not been systematically
addressed. There are always measuring uncertainties in practical conditions, thus one should account for the measuring
errors when evaluating the performance of a data inversion method. Because of the relatively low resolution of the PSM,
the matrix connecting the particle size distribution and the observed total number concentration is usually ill-conditioned.
The kernel function method may theoretically recover the observed particle size distribution when there are no random
errors. However, it sometimes leads to large uncertainties when there are small random errors because of the instability
of the least square method at a near collinear data set (Ellis, 1998).
The equation mapping the particle size distribution to the particle number concentration detected by the PSM is the
Fredholm integral equation of the first kind, which arises in many fields, e.g., when studying the molecular dynamics in



complex systems (Schäfer et al., 1996) and characterizing the transfer function of an ion drift tube (Buckley and Hogan,
2017). Various types of aerosol spectrometers or classifiers, e.g., cascade impactors, optical particle spectrometers,
electrical mobility spectrometers, and diffusional barriers, classify particles according to the signals recorded by a number
of channels. There is no strict one-to-one relationship between the particle number concentration in a certain size range
and the detected signal in a certain channel because of the finite sizing resolutions. The inversion methods used in the
previous aerosol spectrometers can possibly be applied to address the PSM inversion problem. The review of the inversion
methods for aerosol spectrometers can be found in Kandlikar and Ramachandran (1999), Knutson (1999), and
Ramachandran and Cooper (2011).
An inversion method with less prior information on the particle size distribution is preferable for the PSM inversion
problem. It is impossible to obtain a continuous particle size distribution using a finite number of the detected signals
without any constraints, e.g., a known analytical expression to describe the size distribution. Some inversion methods rely
on a presumed particle size distribution formula (Fuchs et al., 1962; Raabe, 1978; Ramachandran and Kandlikar, 1996)
or prior information on the detection efficiencies (e.g., Onischuk et al., 2017). However, approximating various shapes
of the observed sub-3 nm particle size distributions or the PSM detection efficiency curves using a specific formula may
lead to relatively large uncertainties. Some methods are feasible in certain conditions, however, sometimes they are not
convergent or may lead to high-frequency oscillations (Twomey, 1975; Ferri et al., 1989) due to practical random errors.
Some methods use smoothing criterions to deal with the oscillations (Markowski, 1987; Winklmayr et al., 1990), however,
they occasionally report an over-smoothed size distribution because of the relatively low resolution and limited size bins
of the PSM. The Tikhonov regularisation (Tikhonov, 1963) uses a regularisation parameter to determine the balance of
smoothing and the agreement with the PSM recorded data, thus the inverted result may be affected by the method to
determine the regularisation parameter (e.g., Wahba, 1977; Hansen, 1992).
Based on the reasons mentioned above, we chose the H&A linear inversion method (Hagen and Alofs, 1983) and the
expectation-maximization algorithm, and tested the feasibility to apply these methods in the PSM inversion problem. The
H&A method is a linear inversion method used in size distribution multi-charge correction, which has the relatively low
computational expense. The expectation-maximization algorithm is an iterative method based on probability theory
(Dempster et al., 1977), and it was used to reconstruct particle size distributions from diffusion battery data (Maher and
Laird, 1985; Wu et al., 1989).
In this study, we tested the performance of the step-wising method, the kernel function method, the H&A method, and
the expectation-maximization algorithm in PSM inversion. Experiments and Monte Carlo simulations accounting for
random errors were used to evaluate the sizing accuracies and the uncertainties of the particle size distributions recovered



using four inversion methods. The influence of particles larger than 3 nm on the reported sub-3 nm particle size
distributions was discussed. Based on the comparison, the methods with comparatively low uncertainties and high
stabilities were recommended to address the PSM inversion problem.
**2 Theory**
**2.1 PSM measuring theory**
A PSM measures the total particle number concentration of the activated particles. The sampled aerosol flow is mixed
with a high-temperature flow containing saturated diethylene glycol coming from the saturator, and then the mixed flow
passes through a low-temperature growth tube. The particles large than a specific diameter can overcome the Kelvin effect
and grow into larger sizes due to the condensation of super saturated diethylene glycol. The detection efficiency is mainly
determined by the particle diameter and the saturator flow rate. The chemical compositions and charging state may affect
the detection efficiencies (Kangasluoma et al., 2013; Kangasluoma et al., 2016a) and lead to errors in the reported particle
size distributions (Kangasluoma and Kontkanen, 2017), however, we mainly focus on the inversion method in this study
and assume the detection efficiency is only size dependent at a certain saturator flow rate. Since the temperatures in the
saturator and the growth tube are fixed, a higher saturator flow rate leads to a higher super saturation ratio of diethylene
glycol in the growth tube hence higher detection efficiencies (Fig. 1a). See Section 3.1 for the details on how to obtain
the detection efficiency curves. The detected total particle number concentration varies with the varying saturator flow
rate when the particle size distribution keeps unchanged. The relationship between the detected total particle
concentration, $R$, the saturator flow rate, $s$, and the particle size distribution function, $n$, can be expressed in the Fredholm
integral equation of the first kind:

$$R_i = \int_0^{+\infty} \eta\left(s_i, d_p\right) \times n \times \mathrm{d}d_p + \varepsilon_i , \tag{1}$$

where $R_i$ is the number concentration recorded at the i[th] saturator flow rate, $s_i$; $d_p$ is the electrical mobility diameter since
the calibrating particles are classified according to their electrical mobility; $\eta$ is the overall detection efficiency determined
by $s$ and $d_p$, including the detection efficiency and the sampling efficiency; $n$ is the probability density of particle number
concentration (particle size distribution function), $\mathrm{d}N/\mathrm{d}d_p$ and $N$ is the accumulated number concentration of particles
smaller than $d_p$; and $\varepsilon_i$ is the error in the recorded particle concentration at $s_i$.
There are many potential sources of the error, $\varepsilon$. For instance, the uncertainties in the calibrated detection efficiencies, the
systematic errors caused by the non-ideal fitting formula of the detection efficiency curves, the CPC counting
uncertainties, the uncertainties in the super saturation ratio due to fluctuations in the flow rate and temperature, and the



unstable aerosol source will all contribute to the difference between the detected number concentration and the expected
particle concentration assuming there is no error.
As shown in Fig. 1b, the kernel function of the PSM, $K$, is defined as the derivative of the detection efficiency, $\eta$, with
respect to the saturator flow rate, $s$. The area of the kernel function is equal to the difference between the detection
efficiencies at the maximum and minimum saturator flow rates. Here we define $r$ as the derivative of the detected number
concentration, $R$, with respect to $s$. According to Eq. 1, the relationship between $r$ and $s$ is also a Fredholm integral
equation of the first kind:

$$r_{\mathrm{m}} = \int_0^{+\infty} K\left(s_{\mathrm{m}}, d_p\right) \times n \times \mathrm{d}d_p + \varepsilon'_{\mathrm{m}} , \qquad (2)$$

where $r_{\mathrm{m}}$ is the $r$ at the m[th] saturator flow rate, $s_{\mathrm{m}}$; and $\varepsilon'_m$ is the error in $r_{\mathrm{m}}$. Although $r$ is theoretically defined as the
derivative of $R$, practically one can only approximate $r$ using the difference between two adjacent $R_i$ over the increment
in $s_i$ and approximate $s_{\mathrm{m}}$ with the mean value of the two corresponding $s_i$. These approximations also contribute to the
uncertainties, $\varepsilon'_m$ in addition to the aforementioned sources for $\varepsilon_i$.
When using a PSM to determine particle size distributions, the PSM records the varying total particle concentration, $R_i$,
and the corresponding saturator flow rate, $s_i$. The saturator flow rate may vary continuously in the scanning mode or fixed
at different flow rates in the stepping mode. The particle size distributions are recovered using the recorded relationship
between $R_i$ and $s_i$ or the relationship between the approximated $r_{\mathrm{m}}$ and $s_{\mathrm{m}}$.
The size ability of the PSM can be described using the size resolution. Similar to the definition of the sizing resolution of
a differential mobility analyser (DMA, Flagan, 1999) to classify particles according to their electrical mobility, we define
the resolution of a PSM as:

$$\mathrm{Res} = \frac{s^*}{\Delta s} \qquad (3)$$

where $Res$ is the resolution at $s^*$; $s^*$ is the peak saturator flow rate of a kernel function; and $\Delta s$ is the full width at half
maximum of the kernel function peak. A relationship between the saturator flow rate and the electrical mobility diameter
is defined to straightforwardly relate the resolution and the particle diameter. The peak saturator flow rate, $s^*$ is defined
as the corresponding saturator flow rate of the particle diameter. This definition is similar to but different from the
definition using the saturator flow rate at the half maximum detection efficiency in Lehtipalo et al. (2014) and in the
commercialized PSM. The sizing resolution of a PSM can be estimated according to the relationship between $s$ and $d_p$, as
shown in Fig. 2. However, one should especially keep in mind that the PSM does not measure particle diameter because
the relationship between $s$ and $d_p$ is only a definition rather than an intrinsic correlation. A PSM only record the varying





particle concentration against the varying saturator flow rate (as indicated in Eqs. 1 and 2). One can only obtain the
particle diameters via proper data inversion.
**2.2 The step-wising method**
The resolution of the PSM is assumed infinite in the step-wising method. Thus, the integral equation relating $n$ and $r$
collapses into a one-to-one corresponding relationship (Lehtipalo et al., 2014),

$$n_m = \frac{2\left(R_{i+1} - R_i\right)}{\eta\left(s_{max}, d_i\right) + \eta\left(s_{max}, d_{i+1}\right)}$$
(4)

where $n_m$ is the particle size distribution function ($dN/dd_p$) at $d_m$; $d_m$, $d_i$, and $d_{i+1}$ are the corresponding half-maximum cut-
off diameters of $s_m$, $s_i$, and $s_{i+1}$, respectively; and $s_m$ is the mean value of $s_i$ and $s_{i+1}$. The relationship between particle
diameter and the saturator flow rate is determined using the saturator flow rate at the half maximum detection efficiency
(Lehtipalo et al., 2014). The step-wising method does not magnify the relative error in measurement since it is a one-to-
one inversion method. However, the inverted results using the step-wising method are perhaps non-negligibly affected by
the relatively low resolutions of the PSM.
**2.3 The kernel function method**
The kernel function method assumes that the particle size distribution can be approximated using several particle size
bins and the detection efficiencies of particles in each size bin are the same. The mathematical description of this
approximation is:

$$r_m \approx \sum_{j=1}^{J} K\left(s_m, d_j\right) \times n_j \times \Delta d_j, \, J \le I - 1,$$
(5)

where $d_j$ is the representing particle diameter of each size bin; $J$ is the number of $d_j$; $n_j$ is the particle size distribution
function ($dN/dd_p$) at $d_j$; $\Delta d_j$ is the length of each size bin; and $I$ is the number of $R_i$. The symbol of $\approx$ is to emphasize that
Eq. 5 is an approximation even if there are no measuring errors because it approximates the integral with a finite discrete
sum and estimates $r_m$ using the recorded $R_i$. Using a matrix, Eq. 5 can be rewritten as:

$$\overset{1}{r}_{(I-1)\times 1} \approx \mathbf{G}_{(I-1)\times J} \cdot \overset{1}{n}_{J\times 1}, \, J \le I - 1$$
(6)

$$\text{where } \mathbf{G}_{i,j} = K\left(s_i, d_j\right) \times \Delta d_j$$
(7)

The subscriptions in the uppercase of Eq. 6 indicate the dimensions of the matrix and the vectors, while the subscriptions
in the lowercase of Eq. 7 represent the corresponding element. The particle size distribution is obtained via solving Eq. 6
using the non-negative least square method.
**2.4 The Hagen & Alofs method**





The H&A method (Hagen and Alofs, 1983) was proposed to deal with the multi-charging correction problem when using
a DMA. It can also be used to solve the PSM inversion problem. Similar to the kernel function method, a discrete sum is
used to approximate the integral:

$$R_i = \sum_{j=1}^{J} \eta\left(s_i, d_j\right) \times n_j \times \Delta d_j, \, J \,\text{?}\, I \tag{8}$$

We use the symbol of $=$ in Eq. 8 rather than $\approx$ because the H&A method requires a $J$ much larger than $I$. One should
increase $J$ if the error in approximating the integral with the discrete sum is still large. Usually, $J$ is determined as 30
times that of $I$ considering the computational expenses. However, Eq. 8 itself is not solvable because there are more
unknown variables than the equations. Thus, additional constraints are required. The H&A method assumes that any $n_j$
can be approximated using $n_i$, i.e.,

$$n_j \approx f\left(\overset{\mathbf{u}}{n_i}, d_j\right), \tag{9}$$

where $f$ is the function relating $n_j$ and $\overset{\mathbf{1}}{n_i}$; $n_i$ is the particle size distribution function at $d_i$; and the vector symbol indicates
that $n_j$ is estimated using more than one single $n_i$. The determination of $d_i$ is theoretically arbitrary as long as the number
of $d_i$ is the same as the number of $R_i$. For the details to determine $f$, please refer to Hagen and Alofs (1983).
Similar to the kernel function method, the relationship between the particle size distribution and the number concentration
recorded by the PSM can be described in the vector form:

$$\overset{\mathbf{u}}{R}_{I \times 1} \approx \mathbf{Q}_{I \times I} \cdot \overset{\mathbf{1}}{n}_{I \times 1} \tag{10}$$

where $\mathbf{Q}$ is determined by $\eta$, $f$, and $\Delta d_j$. One can directly solve Eq. 10 (e.g., via Gaussian elimination) since $\mathbf{Q}$ is usually
non-singular. However, different from the matrix obtained from a DMA, the matrix $\mathbf{Q}$ in PSM inversion problem is
usually not a positive-definite matrix because the detected particle concentration sometimes decreases with the increasing
saturator flow rate due to random errors. Simply solving Eq (10) often obtains negative values in particle size distributions.
Thus, the non-negative least square method is suggested to determine the particle size distribution in the PSM inversion
problem. The H&A methods can also reconstruct the particle size distribution according to the relationship between $r_m$
and $s_m$. However, using the kernel functions instead of the detection efficiencies does not necessarily improve the accuracy
or precision of the results. On the contrast, we found that using the kernel functions usually lead to larger uncertainties
than using the detection efficiencies because of the errors caused by approximating $r_m$.
The H&A method is theoretically more stable than the kernel function method because of the more accurate assumption
of the true aerosol size distribution. However, the H&A method adapted for PSM inversion may still report size
distributions with large uncertainties because of using the least square method. The computational expense of the H&A





method is similar to that of the kernel function method because the rate-limiting step is to solve the least square question.
Their low computational expense is an advantage over other nonlinear inversion methods.
**2.5 The expectation-maximization algorithm**
The EM algorithm is a statistical method dealing with inversion problems with unobserved latent variables. An
explanation of the EM algorithm can be found in Do and Batzoglou (2008). In the PSM inversion problem, the latent
variable is $R_{i,j}$, defined as the contribution of particles with the diameter of $d_j$ to the detected number concentration, $R_i$
(Maher and Laird, 1985). The algorithm obtains the recovered particle size distribution using two steps: the expectation
step and the maximization step. In the expectation step, the values of $R_{i,j}$ are estimated according to Bayesian theorem:

$$R_{i,j} = \frac{n_j \times \eta\left(s_i, d_j\right) \times \Delta d_j}{\sum\limits_{j=1}^{J} n_j \times \eta\left(s_i, d_j\right) \times \Delta d_j} \tag{11}$$

In the maximization step, the particle size distribution function is estimated according to the maximum likelihood:

$$n_j = \frac{\sum\limits_{i=1}^{I} R_{i,j}}{\sum\limits_{I=1}^{I} \eta\left(s_i, d_j\right) \times \Delta d_j} \tag{12}$$

The EM algorithm obtains the recovered particle size distribution by repeating the expectation step and the maximization
step until convergence. The convergence can be measured by the likelihood function (Maher and Laird, 1985). The values
and the number of $d_j$ are not limited when using the EM algorithm, and a larger $J$ can reduce the errors in approximating
the integral using the discrete sum. Thus, the EM algorithm is able to report particle size distributions with more size bins
compared to the step-wising method, the kernel function method, and the H&A method.
The EM algorithm is more stable compared to the algorithms based on the least square methods (Maher and Laird, 1985).
The convergence of the EM algorithm has been proved (Dempster et al., 1977), however, the convergence speed is not
mathematically guaranteed. Compared to the kernel function method and the H&A method, the computational expense
of the EM algorithm is much higher. In addition, the EM algorithm is a greedy algorithm such that the iteration is easily
trapped in a local optimum. To start the first expectation step, an initial guess of the particle size distribution is required.
We suggest the initial guess to be a vector of all ones. Note that the EM algorithm is sensitive to the initial guess and
using a recovered particle size distribution obtained from another method, e.g., the step-wising method does not
necessarily improve the iteration results.



**3 Methods**
**3.1 Experiments**
Laboratory experiments using particles with known peak size or size distribution were conducted to test the inversion
methods (Fig. 3). Sub-10 nm tungsten oxide particles were generated using a wire generator (Peineke et al., 2006;
Kangasluoma et al., 2015). In the narrow peak measurement, the negatively charged particles were classified using a
high-resolution Herrmann DMA. The sizing resolutions of the Herrmann DMA in the experimental conditions were no
smaller than 25 (Kangasluoma et al., 2016b). Thus, the classified aerosols out of the Herrmann DMA can be
approximately regarded as monodisperse. The relationship between the Herrmann DMA voltage and the classified particle
size was calibrated using standard molecular ions (Ude and de la Mora, 2005). A TSI 3068B aerosol electrometer using
the same aerosol flow rate with the PSM (2.5 liters per minute, lpm) was used as the reference.
In the wide peak measurement, the particle size distributions classified using a TSI nanoDMA have wider peaks than
those generated in the narrow peak measurement. The aerosol and sheath flow rates of the nanoDMA were 2 and 10 lpm,
respectively. It should be clarified that the particle size distribution classified using the nanoDMA in the wide peak
measurement were still narrow due to the limitation of the nanoDMA. A lower sizing resolution either achieved by a
higher aerosol-to-sheath flow ratio will cause the nanoDMA out of work due to significant turbulence. A half-mini DMA
(Fernández de la Mora and Kozlowski, 2013) with calibrated penetration efficiency and a downstream Faraday cage
electrometer (FCE) were used to measure the classified particle size distributions in parallel.
The PSM (Airmodus A11) was calibrated using negatively charged tungsten oxide particles before the test. The
experimental setup for the calibration was the same with that used in the narrow peak measurement. The influence of the
finite resolution of the Herrmann DMA on the calibrated efficiency curves was negligible. The saturator flow rate of the
tested PSM varied from 0.05 to 1.3 lpm. This saturator flow rate range is wider than that of a typical PSM to obtain a
complete kernel function curve of 3 nm particles. The maximum background noise of the PSM was approximately 1
No./cm$^3$, which was negligible compared to the usually detected particle concentrations. The detection efficiency is
determined as the ratio of the particle number concentrations reported by PSM over the number concentration reported
by the electrometer. The detection efficiency curves of the PSM were fitted using a function (Eq. 13) modified from the
Chapman-Richards growth curve (Richards, 1959) which fitted better than other tested functions for the tested PSM,

$$\eta = a \times \left[1 + |b| \times (s - s_{max})\right] \times \left[1 - \exp(-c \times s)\right]^{d},$$
(13)

where $s_{max}$ is the maximum saturator flow rate (1.3 lpm); $a$, $b$, $c$, and $d$ are the fitting parameters. If not specially
mentioned, the PSM was fixed at 18 different saturator flow rates when measuring the particle size distributions in this
study. This operation in the stepping mode was to avoid the potential uncertainties introduced in the scanning mode. The



stability of the particle size distribution was monitored using the reference FCE during the relatively long measuring
period.
**3.2 Simulation**
The performance of the four inversion methods was also studied using Monte Carlo simulations. The detection efficiencies
used in the simulations were determined according to the calibrated efficiencies but slightly adjusted towards smoother
curves. The uncertainties in practical calibration were neglected in the simulation.
The particle number concentrations detected at different saturator flow rates were simulated using a certain initial particle
size distribution. The random error, $\varepsilon_i$, was inserted into the simulated particle concentration, $R_i$. The random errors were
determined experimentally. The relative random errors were larger than the statistical relative errors predicted using
Poisson distribution (Iida, 2008; Kuang et al., 2012; Kangasluoma and Kontkanen, 2017) and independent of the particle
concentrations at a certain instrumental configuration, indicating that random errors were governed by the fluctuations of
the source and/or the instrumental parameters (e.g., flow rate). We used the mean relative random standard deviation
observed in the experimental tests, 3.7%, as the representative value. Totally 10 data points were assumed to be collected
at each saturator flow rate. Thus, the random errors inserted into the simulated particle concentrations, i.e., the relative
standard deviations of the mean particles concentrations, were assumed to be 1.2% ($=3.7\%/\sqrt{10}$). A relatively large
random error of 10% obtained from the ambient measurements was also tested. The Monte Carol simulation was
conducted for 10000 times using each inversion method to estimate the accuracy and precision of the recovered particle
size distribution indicated by the mean values and the standard deviations of inverted results.
**4 Results and discussion**
**4.1 Sizing accuracy**
The inversion methods tested in this study, i.e., the step-wising method, the kernel function method, the H&A method,
and the EM algorithm are able to estimate the classified particle diameters when the PSM was measuring nearly
monodisperse sub-3 nm particles. When the classified particle diameters were 1.51 nm and 2.41 nm, respectively, all of
the four inversion methods can recover single peaks around the classified diameter (Figs 4a, 4b). The size distribution
reported by the step-wising method was the widest because the step-wising method does not account for the resolution of
the PSM. Note that the peak diameters reported by the kernel function method and the H&A method were also affected
by the selection of the particle size bins. The total particle concentrations obtained via inversion were similar to the
number concentration detected by the reference FCE, except for the number concentration of 1.51 nm particles reported
by the kernel function method.



None of the four inversion methods could size particles larger than 3 nm with relatively good sizing accuracies. When
the classified particle diameter was 3.93 nm, the four inversion methods failed to report narrow peaks with peak diameters
approximating 3.93 nm (Fig. 4c). This is because the PSM resolution for particles larger than 3 nm is low, i.e., the
resolution was ~1.0 when measuring the classified 3.93 nm particles (Fig. 2). When focusing on the sub-3 nm particle
size range, the kernel function method, the H&A method, and the EM algorithm reported nearly no sub-3nm particles.
However, the step-wising method reported a non-negligible amount of sub-3 nm particles with a total number
concentration of 1591 No./cm$^3$ due to the low sizing resolution.
We further tested the sizing ability of the four inversion methods using the sum of the recorded particle concentrations
when the PSM was measuring 1.51, 2.41, and 3.93 nm particles (Fig. 4d). The kernel function method, the H&A method,
and the EM algorithm distinguished the particles with different sizes, and the reconstructed peaks were similar to the
corresponding peaks when the PSM was measuring monodisperse particles. The inverted results using the step-wising
method was also unaffected by the summation, however, it was difficult to distinguish the isolated peaks from the
recovered particle size distribution due to the broadened size distribution.
The size distributions of particles larger than 3 nm could not be successfully retrieved via data inversion because of the
low resolution of PSM for these particles, however, it helped to recover sub-3 nm particle size distributions. Most of the
reported particle sizes using the kernel function method, the H&A method, and the EM algorithm were larger than 3 nm
when the PSM was measuring 3.93 nm particles (Fig. 4c). This estimation of particles larger than 3 nm assured a relatively
accurate sizing of sub-3 nm particle size distribution (Fig. 4d). Thus, we recovered the particle size distribution up to 5
nm using different inversion methods but focus only on the sub-3 nm size range.
**4.2 Uncertainties using different inversion methods**
The step-wising method, the kernel function method, and the H&A method may report false sub-3 nm particles when
there are only particles are larger than 3 nm in the input aerosol. A particle size distribution with a peak diameter of 5 nm
and nearly no sub-3 nm particles was simulated (Fig. 5a). The detected particle concentrations were assumed to fluctuate
with a 1.2% relative standard deviation due to measuring uncertainties (Fig. 5b). The EM algorithm reported nearly no
sub-3 nm particles except for the smallest size bin at 1.16 nm (Fig. 5c). The expected values of particle concentrations in
the bins smaller than 3 nm recovered using the H&A method were near zero, however, false sub-3 nm particle
concentrations were occasionally reported (Fig. 5d). Compared to the H&A method, the size distribution recovered using
the kernel function method was more unstable, especially in the sub-2 nm size range (Fig. 5e). Different from the H&A
method and the kernel function method that reported false results due to their instability, the step-wising method reported
false particle size distributions with nonzero mean values (Fig. 5f). This is because the step-wising method assumes a



simple one-to-one relationship between the saturator flow rate and the recovered particle diameter instead of accounting
for the wide kernel function peaks.
The false sub-3 nm particle concentrations due to improper inversion methods were tested experimentally. Particles larger
than 5 nm were classified using the nanoDMA (Fig. 6a). No sub-3 nm particles were reported using the EM algorithm
and the H&A method. On the contrast, the kernel function method and the step-wising method reported approximately
$3 \times 10^3$ particles when the total particle concentration measured using the DMA-FCE system was approximately $2.4 \times 10^4$.
Based on both the simulating and experimental results, we conclude that the PSM may report false sub-3 nm particle size
distributions when there are actually no sub-3 nm particles because of using non-ideal data inversion methods, especially
the step-wising method. Note that large particles whose detection efficiencies do not vary with the saturator flow rate do
not lead to a bias in the recovered sub-3 nm particle concentrations. We examined this theoretical deduction
experimentally using a PSM to measure ambient particles existing in the room air and the recorded particle concentration
did not significantly vary with the saturator flow rate.
The performance of the four inversion methods in the sub-3 nm size range under the influences of larger particles was
tested using a bimodal distribution (Fig. 7a). Similar particle size distributions are usually observed in the atmospheric
new particle formation events (Jiang et al., 2011) and in flame (Tang et al., 2017). As shown in Fig. 7, the particle size
distribution recovered using the EM algorithm had the highest accuracy and the smallest uncertainties among the four
methods. The recovered particle size distribution using the EM algorithm had a slightly different shape compared to the
initial distribution because the results were trapped in the local optimum. However, the differences between the recovered
and the initial size distributions were the smallest. The standard deviations of the size distribution recovered using the
H&A method and the kernel function method were relatively large due to the unstable least square method. Because of a
better assumption of the initial particle size distribution, the H&A method resulted in smaller uncertainties compared to
the kernel function method, especially in the sub-2 nm size range. The size distribution recovered using the EM algorithm
has higher accuracy and stability compared to both the H&A method and the kernel method because the one-to-one
inversion method does not magnify relative errors.
The experimental tests using bimodal distributions agreed with the simulation results. The particles with a peak diameter
at approximately 2.3 nm were classified using the nanoDMA. We added the observed number concentration to those
detected in Fig. 6a (particles larger than 5 nm) to account for the influence of large particles. Unfiltered room air served
as the makeup flow to provide background particles. As shown in Fig. 8, all the four inversion methods recovered the
peak around 2.3 nm, while the results reported by the H&A method and the kernel function method were less smooth
compared to the EM algorithm and the step-wising method.





Smoothing the size distribution recovered using the H&A method and the kernel function method into fewer size bins
can reduce the uncertainties. We determined the number of the size bins of the recovered distributions according to the
number of the fixed saturator flow rates. Too many size bins will lead to relatively large uncertainties, however, the
uncertainties can be reduced by sacrificing the resolution, i.e., reporting the size distribution in fewer bins. The size
distributions recovered using the kernel function method were reported in typical 4-6 bins (Lehtipalo et al., 2014). This
was achieved by assuming fewer discrete particle diameters in Eq. 5. Another option is to merge bins into fewer numbers
after inversion rather than assume fewer bins at the beginning. Note that the H&A method cannot assume fewer discrete
size bins at the beginning. Instead, the H&A method assumes an adequate number of size bins to guarantee a relatively
smooth distribution (Eq. 8). As shown in Fig. 9, the standard deviations of the reported size distribution with fewer size
bins were comparatively smaller than the corresponding standard deviations with more size bins shown in Fig. 7. The
H&A method reported size distributions with smaller standard deviations than the kernel function method, and the kernel
function reported in merged size bins had smaller standard deviations than the kernel function method using fewer size
bins at the beginning. This is because approximating the true particle size distribution, which is usually a smooth curve,
with fewer discrete size bins will lead to larger uncertainties. Thus, we suggest merging the recovered particle size
distribution into a few size bins to reduce the uncertainties when using the H&A method and the kernel function method.
Relatively large uncertainties were found when recovering sub-1.3 nm particle size distributions. A particle size
distribution with an increasing $dN/dd_p$ as a function of the decreasing particle diameter, which is a typical particle size
distribution observed in the atmospheric new particle formation events (Jiang et al., 2011), was used to test the four
inversion methods (Fig. 10). None of the inversion methods reported a particle size distribution with relatively small
uncertainties comparable to the inverted results shown in Fig. 7c, especially in the sub-1.3 nm size range. Similar to the
results for particles larger than 3 nm, the low resolution of particles smaller than 1.3 nm (Fig. 2) is possibly the cause of
the large uncertainties. In addition, incomplete kernel function peaks and the relatively low detection efficiencies of sub-
1.3 nm particles may also contribute to the uncertainties (Fig. 1).
The performance of the inversion methods under relatively large random errors was also tested. The relative standard
deviation used in the above simulations, 3.7%, was estimated according to laboratory experiments. The relative standard
deviations of the recorded particle number concentration obtained from the atmospheric measurement were usually
similar to the value obtained in the laboratory, indicating the random errors were governed by instrumental factors.
However, relatively large uncertainties in the recorded particle number concentrations were sometimes observed due to
the unstable atmospheric aerosol source. Thus, we simulated the performance of the four inversion methods using a
relative standard deviation of 10%. It should be clarified that the value 10% only characterizes the random errors of the





CPC since it was estimated using the data when the recorded particle number concentration did not vary with the saturator
flow rate. Compared to the results in Fig.7 simulated using the same aerosol size distribution, the uncertainties in the
recovered particle size distributions using the larger relative standard deviation of 10% was larger (Fig. 10). The EM
algorithm still reported smaller uncertainties compared to the H&A method and the kernel function method. Note the
expected value of sub-2 nm particle size distribution recovered using the kernel method was close to the input size
distribution when the uncertainty was 3.7% (Fig. 7); however, the recovered size distribution in the sub-2 nm size range
was non-negligibly overestimated when the uncertainty was 10% (Fig. 10).
**4.3 Uncertainties in the scanning mode**
The PSM instrumental factors limiting the accuracy of the inversion were also tested. Although using the EM algorithm
and the H&A method can reduce the errors of the recovered size distributions compared to the kernel function method
and the step-wising method, relatively small measuring uncertainties are still vital to retrieve a particle size distribution
with relatively high accuracies. The uncertainties in the scanning mode, for example, is one of the potential sources of
the measuring uncertainties. The saturator flow rate of a scanning PSM increases linearly with time in previous studies.
However, the relationship between the particle diameters and the saturator flow rates at the kernel function peaks is
nonlinear (Fig. 2). The detection efficiencies of particles larger than 1.6 nm vary mainly in the flow rate range from 0.05
to 0.3 lpm while the corresponding scanning time is only 20% of the whole scanning cycle. This nonlinear relationship
may result in non-negligible uncertainties in the recovered particle size distributions (Fig. 12). The EM algorithm
recovered the single peak when using the particle concentrations recorded in the stepping mode. However, the recovered
particle size distribution using the EM algorithm was not a single smooth peak when using data recorded in the scanning
mode (Fig. 12). This difference can be illustrated using the raw data. The curves of the particle number concentration
recorded in the stepping mode and the scanning mode are similar to each other and they both appear to be smooth (Fig.
13a). When presenting in the derivate of the particle number concentration with the respect to saturator flow rate, however,
the curve corresponding to the stepping mode appeared to be a single peak while the other curve corresponding to the
scanning mode seemed to be composed of multiple single peaks (Fig. 13b). Since none of the four inversion methods
tested in this study add smoothing constraints when solving the Fredholm integral equation of the first kind, this roughness
in the raw data will lead to split peaks in the recovered particle size distribution unless one report the size distribution
using only a few size bins.



### 4.4 Implications on using the PSM

According to the discussion above, we provide the following suggestions on using a PSM to determine particle size distributions:

(a) Particle size range and saturator flow rate range. Complete efficiency curves are preferable to determine the particle size distribution in a certain size range. For example, to reduce the uncertainties in the recovered size distribution of ~3 nm particles, the saturator flow rate in this study was extended from the commonly used 0.1 lpm to 0.05 lpm where the detection efficiency of 3.11 nm particles was almost zero. The detection efficiency curves of particles larger than the maximum concerned diameter should also be calibrated to reduce the influence of large particles on the recovered particle size distribution and total concentration. The PSM can theoretically estimate particle size distributions larger than 3 nm or smaller than 1.3 nm, however, the uncertainties are usually large due to the low resolution and the incomplete detection efficiency curves. The particles whose detection efficiency are constant values in the measuring saturator flow rate range cannot be determined using a PSM and they do not influence the recovered particle size distributions if their concentrations are sable during each scanning cycle.

(b) Scanning scheme. The scanning scheme of the saturator flow rate is suggested to be improved to reduce the measuring uncertainties. The scanning scheme is preferably determined to ensure that the particle diameter corresponding to the saturator flow rate increases linearly with time so that the numbers of the recorded particle number concentration at each saturator flow rate are the same when the recovered particle size increases linearly. A convex function between the saturator flow rate and the scanning time, e.g., an exponentially increasing saturator flow rate, is also better than the linear scanning scheme. Such improvement may require updating both the hardware and the software.

(c) Inversion method. We suggest using the EM algorithm to address the PSM inversion problem because the particle size distributions recovered using the EM algorithm have the best accuracy and stability among the four tested methods. However, considering the relatively high computational expense of the EM algorithm, the H&A method reporting in merged size bins is recommended to be used for preliminary data analysis and for meeting the need of fast inversion, e.g., real-time display on the instrumental screen. The accuracy of the recovered size distribution is also determined by the uncertainties in the recorded number concentration rather than the inversion method alone. The inversion methods suggested in this study does not necessarily assure an accurate inverted result without properly determined detection efficiencies and an improved scanning scheme.

(d) Uncertainties in atmospheric measurement. One should be always aware of the potential uncertainties in the recovered particle size distribution, especially when conducting atmospheric measurement. The reported sub-3 nm particle concentrations may be false results due to systematic and random error, especially when using the step-wising method.



The number of the reported size bins should also be carefully limited. For example, the EM algorithm can theoretically
provide infinite size bins; however, we suggest reducing the reported size bins to avoid false fluctuations.
**5 Conclusions**
We tested the performance of four inversion methods to recover particle size distributions from the particle size magnifier
data using Monte Carlo simulation and experiments. The four inversion methods are the step-wising method, the kernel
function method, the H&A method, and the EM algorithm, respectively. The step-wising method may report false sub-3
nm particle concentrations when there are no sub-3 nm particles in the input aerosol because it does not account for the
influence of particles large than 3 nm. The kernel function method and the H&A method may lead to relatively large
uncertainties in the recovered particle size distribution because of using the unstable least square method, and they
occasionally report false sub-3 nm concentrations due to the large uncertainties. Compared to the kernel function method,
the H&A lead to smaller uncertainties while having a similar computation expense. This is because that the H&A method
assumes a near continuous size distribution rather than a discrete distribution with limited size bins. One can reduce the
uncertainties via merging the particle size distribution reported by the H&A method into fewer size bins. Among the
tested inversion methods, the EM algorithm has the highest accuracy and stability. Another advantage of the EM algorithm
over the other three methods is that it does not limit the number of the particle size bins. The instrumental factors also
limit the accuracy and precision of the recovered particle size distribution. The uncertainties of the recovered size
distributions of particle smaller than 1.3 nm or larger than 3 nm may be significant due to the incomplete kernel function
curves, the low resolution and/or the low detection efficiency. The measuring uncertainties in the scanning mode may
also increase the uncertainties of the recovered size distribution.
Based on this study, we suggest that a) the EM algorithm is used to recover the particle size distribution measured by the
PSM and the H&A method can be used for preliminary data analysis and for fast inversion purposes; b) the hardware and
software of the PSM should be improved to reduce the measuring uncertainties, e.g., via changing the scanning scheme
of the saturator flow rate; c) one should carefully distinguish the false inverted results from the true sub-3 nm particles,
especially in the sub-2 nm size range and/or when using the step-wising method.
**Data availability**
The characterizations of the tested PSM are shown in the figures. The Matlab scripts for the inversion methods are
available upon request.



## 1 Competing interests

The authors declare that they have no conflict of interest.

## 3 Acknowledgement

Financial supports from the National Key R&D Program of China (2017YFC0209503), the National Natural Science Foundation of China (21521064 & 41730106), ACTRIS-2 (grant agreement No. 654109), the Academy of Finland (project No. 307331), and Faculty of Science, University of Helsinki, are acknowledged. R. Cai appreciates the support from China Scholarship Council (CSC) for his visit to University of Helsinki.

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



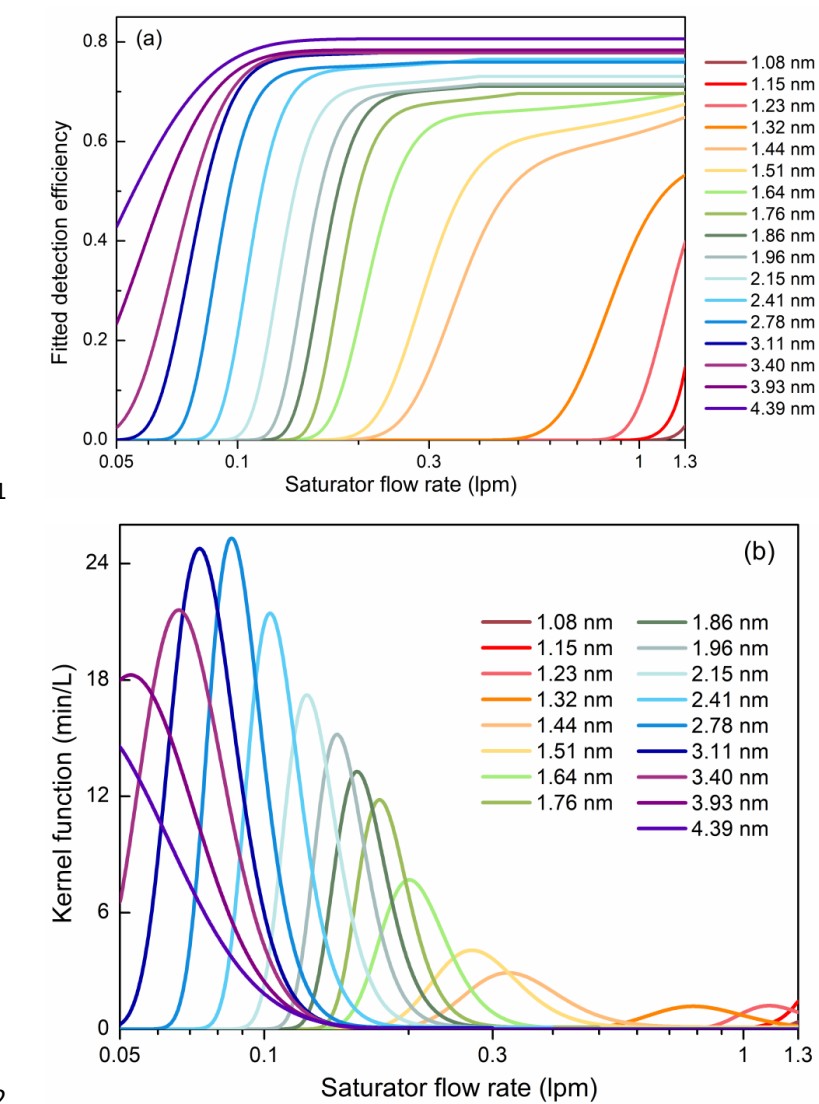

**Figure 1** (a) The fitted detection efficiency curves according to calibration data. (b) The estimated kernel function curves according to

the fitted detection efficiencies. The kernel function is equal to the derivative of the detection efficiency with the respect to the saturator

flow rate.





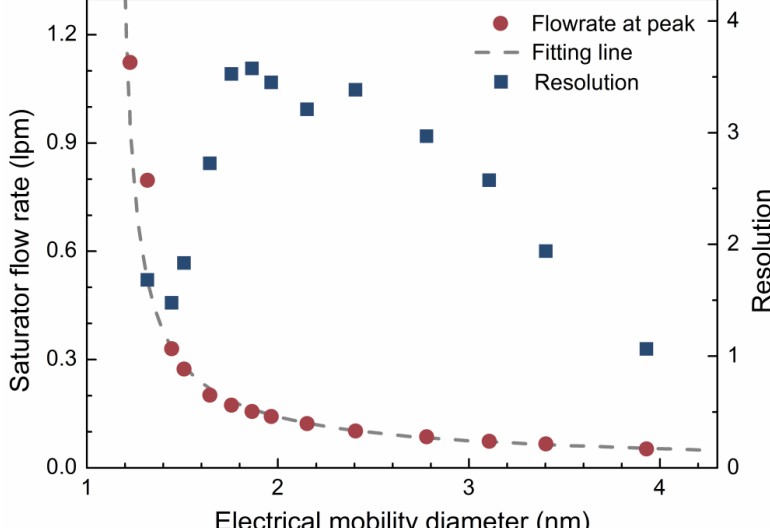

2   **Figure 2** The saturator flow rate at kernel function peak and the resolution as functions of the particle diameter. Note that the resolution

3   is defined using the saturator flow rate, however, the horizontal axis is shown in the particle diameter corresponding to the peak

4   saturator flow rate for more straightforward understanding.



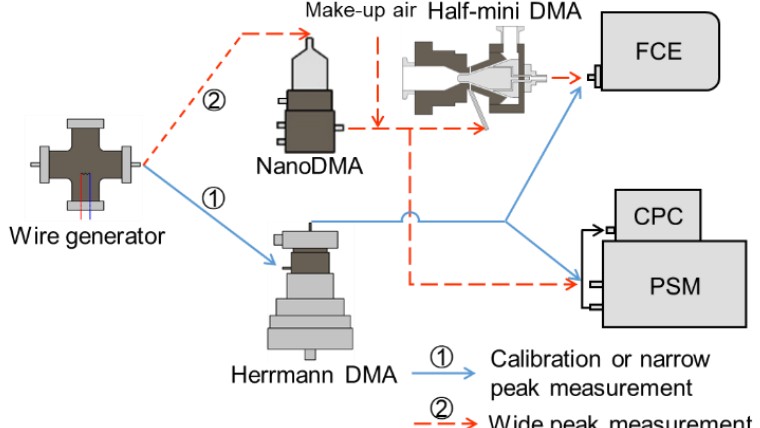

2      **Figure 3** The experimental setup to calibration the PSM and test the inversion methods.
3





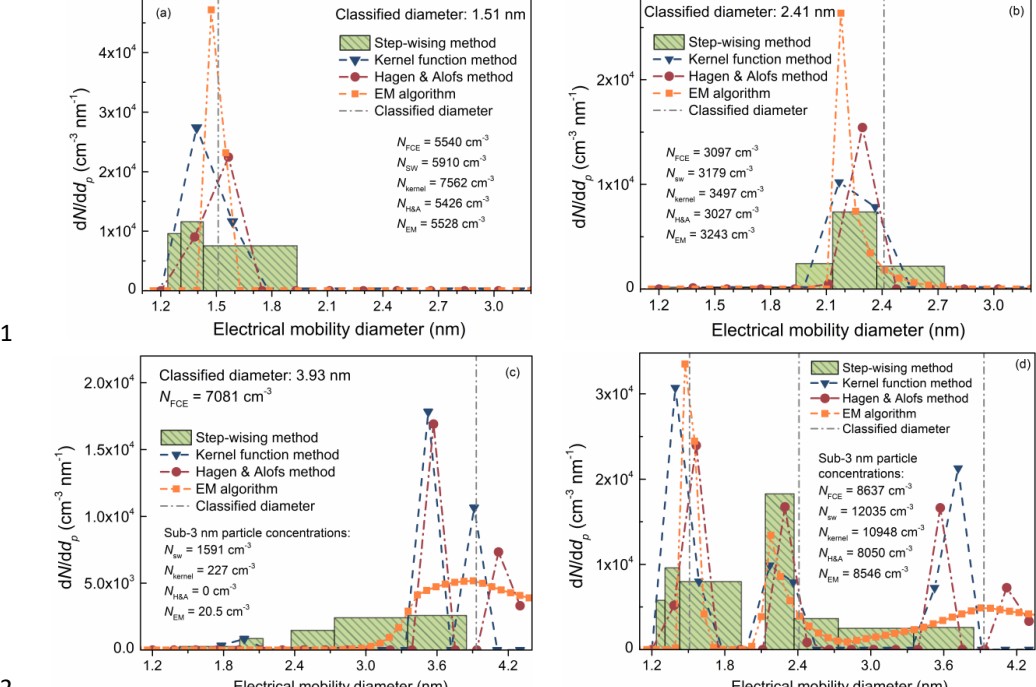

**Figure 4** The recovered particle size distributions using different inversion methods when measuring monodisperse particles. FCE, SW, kernel, H&A, and EM are short for the Faraday cage electrometer, the step-wising method, the kernel function method, the H&A method, and the expectation-maximization algorithm, respectively. The number concentration detected by the reference FCE and the sum of recovered sub-3 nm particle concentration in each size bin are shown in the text. The size distributions in (d) were recovered using the sum of the recorded number concentrations in (a), (b), and (c), i.e., assuming the PSM was measuring 1.51, 2.41, and 3.93 nm particles simultaneously.





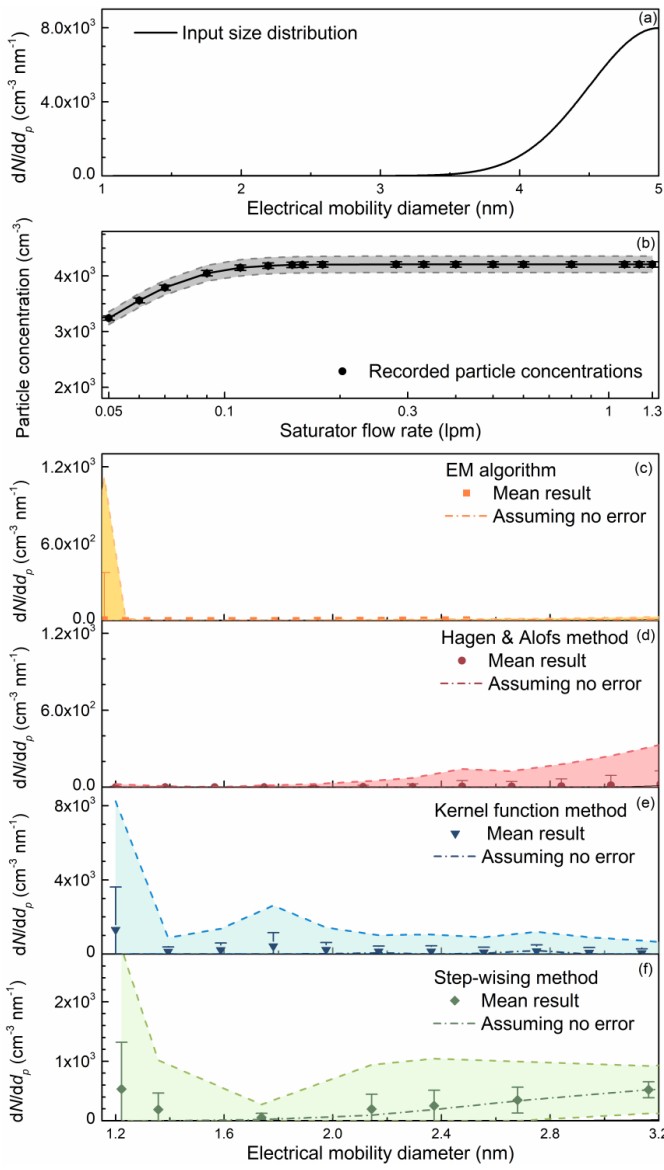

**Figure 5** The recovered sub-3 nm particle size distributions simulated using the Monte Carlo method when the detected particles were
larger than 3 nm. (a) The assumed true particle size distribution. (b) The simulated particle concentrations recorded by the PSM. The
concentrations were assumed to fluctuate due to random errors. The particle size distributions were recovered using (c) the EM
algorithm, (d) the H&A method, (e) the kernel function method, and (f) the step-wising method. The error bar represents the standard
deviation of the recorded particle concentration or the recovered size distribution, and the shaded area indicates the range determined
by three times the standard deviation. The dashed lines represent the inverted results assuming there were no random errors in the
recorded particle number concentrations. Note that the scale of the vertical axis in (c-f) is different and the appearing possibility of
recorded counts or the recovered size distribution is not uniform in the shaded area.





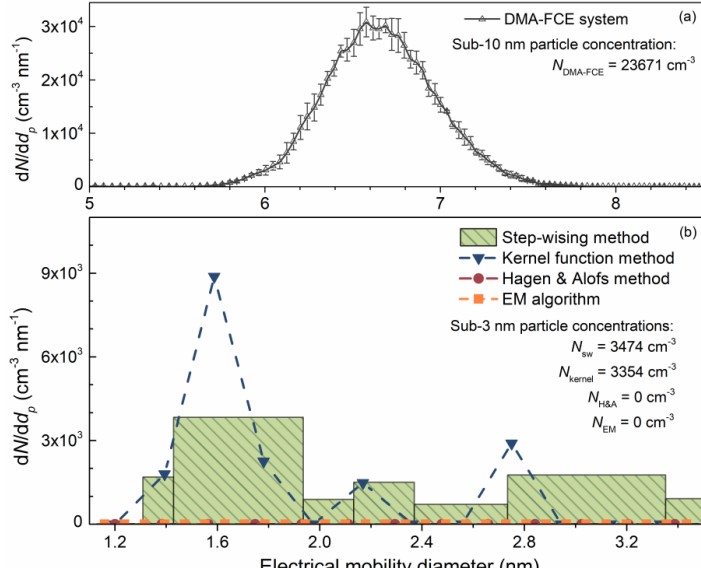

2  **Figure 6** The experimental testing results of the four inversion methods when the PSM was measuring particles larger than 3 nm. (a)

3  The particle size distribution detected by the reference halfmini DMA-FCE system. (b) The recovered particle distributions using

4  different inversion method.



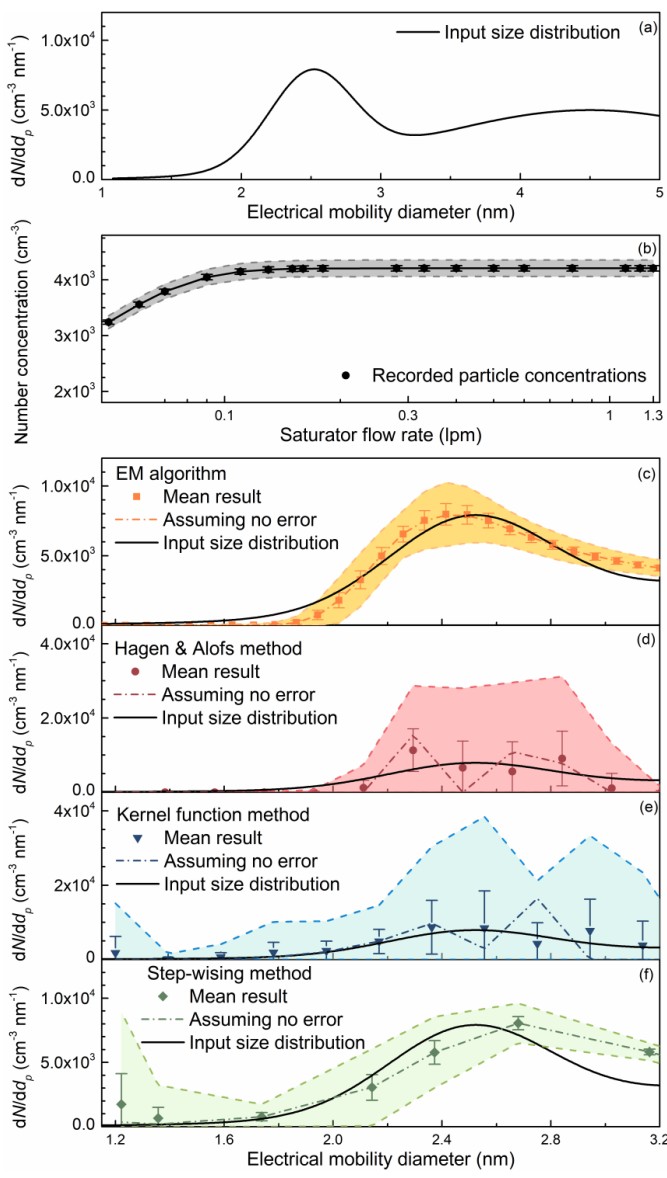

2    **Figure 7** The input and recovered sub-3 nm particle size distributions simulated using the Monte Carlo method. Note the vertical axes
3    in panel (c-f) are not the same.





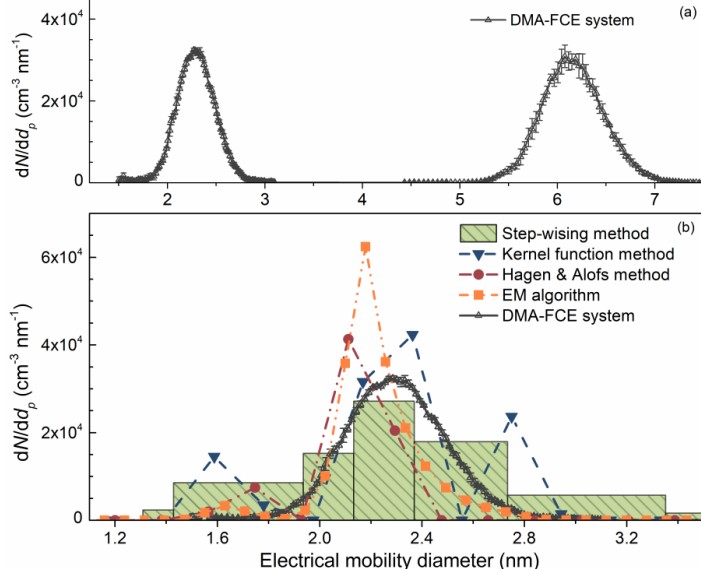

**Figure 8** The experimental testing results of the four inversion methods when the PSM was measuring sub-3 nm particles with the influence of larger particles. The particle number concentrations for inversion and the particle size distribution detected using the DMA-FCE system were the sums of two separate experiments rather than real data obtained in a single experiment.



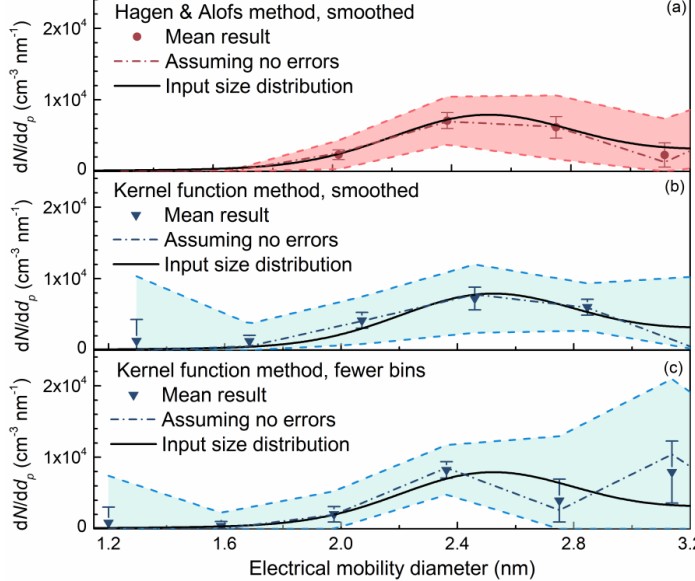

2 **Figure 9** Comparisons of the inverted results using (a) the H&A method smoothing the particle size distribution via merging size bins;

3 (b) the kernel function method smoothing the particle size distribution via merging size bins; and (c) the kernel function method

4 assuming fewer discrete particle sizes in Eq. 5.

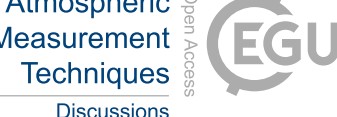



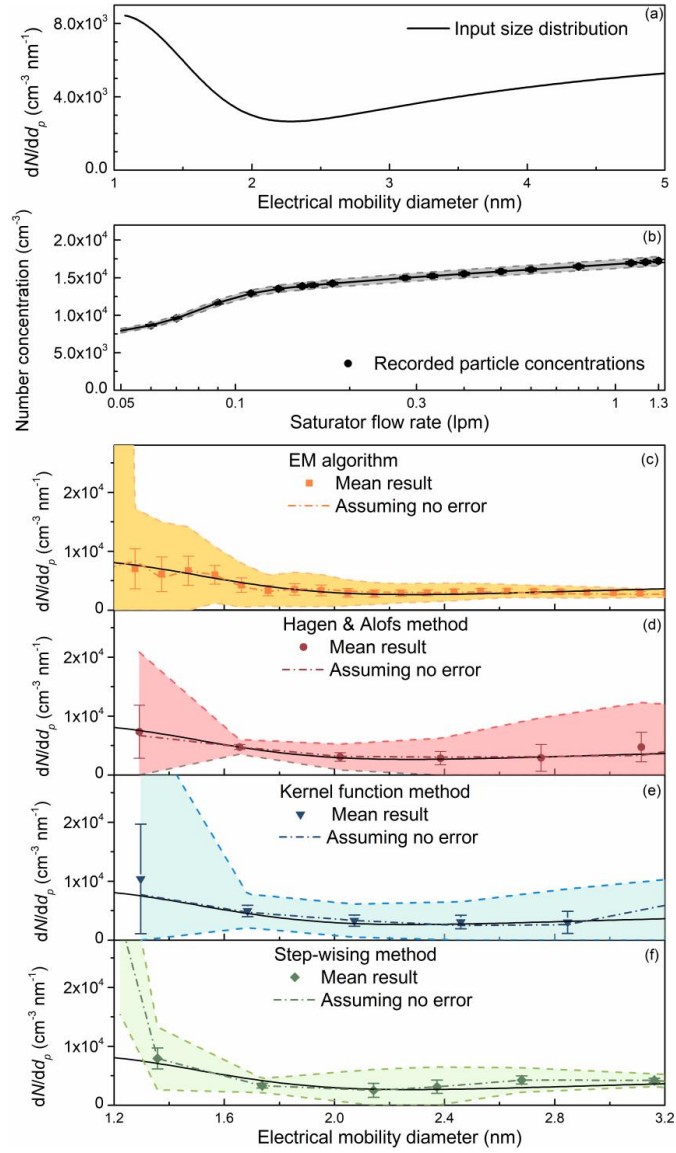

2 **Figure 10** The input and recovered sub-3 nm particle size distributions simulated using the Monte Carlo method when the particle size

3 distribution increases with the decreasing particle diameter.

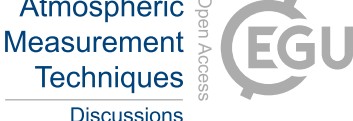

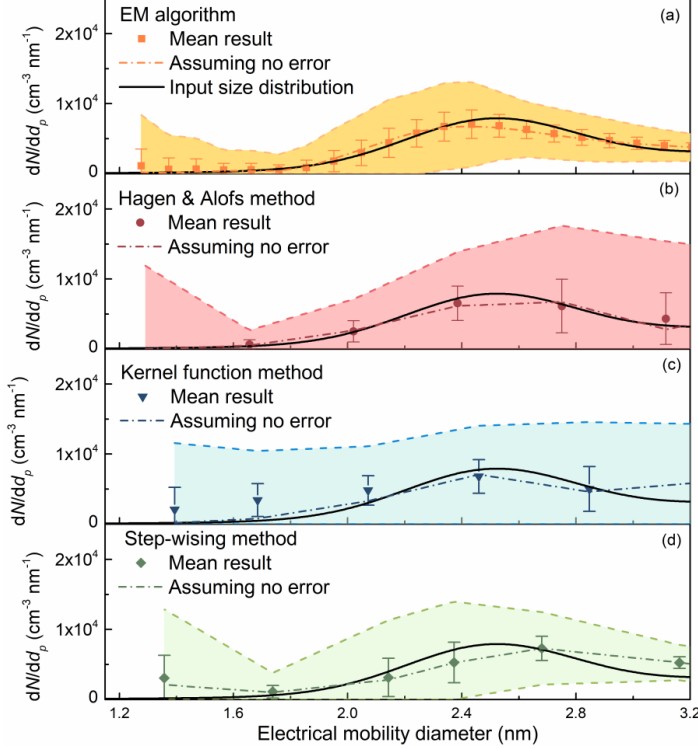

2 **Figure 11** The recovered particle size distributions simulated using the Monte Carlo method when assuming the relative standard

3 deviation of the recorded particle number concentration is 10%. The reported size bins smaller than 1.3 nm recovered using the kernel

4 function method and the step-wising method are not shown because of the large uncertainties.



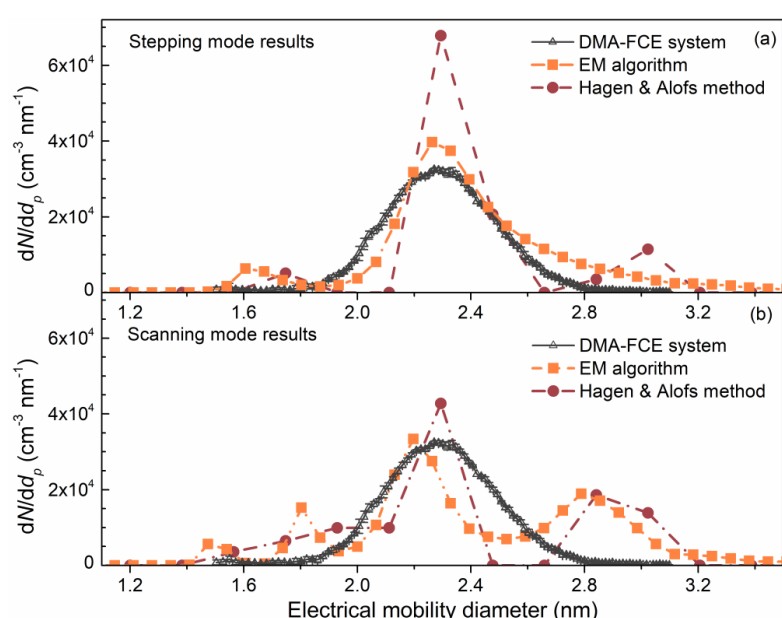

**Figure 12** The recovered particle size distributions using the particle number concentration recorded in (a) the stepping mode and (b)

the scanning mode.



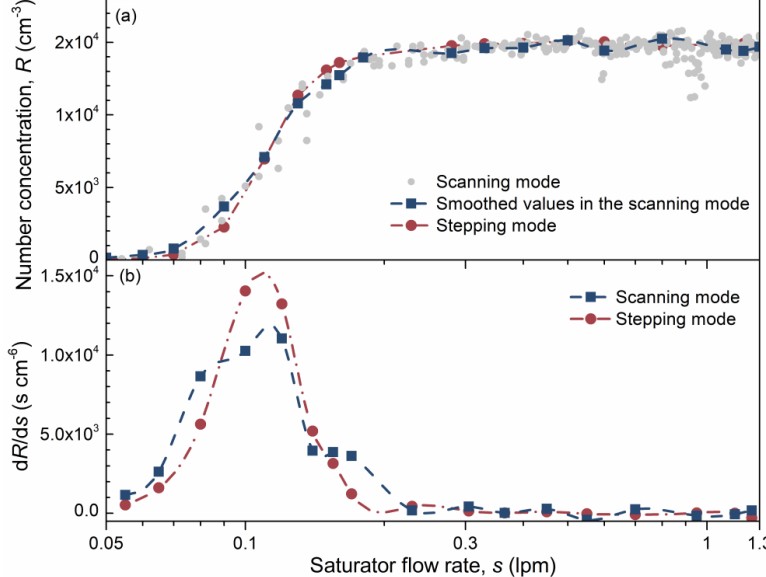

2 **Figure 13** (a) The relationship between the recorded particle number concentration and the saturator flow rate in the scanning mode

3 and the stepping mode. (b) The derivative of number concentration with the respect to the saturator flow rate.

