# Peer review of "Data inversion methods to determine sub-3 nm aerosol size"

_Atmospheric Measurement Techniques, 2018_

## Referee Comment (RC1) · Anonymous Referee #1 · 2 May 2018

General comments:

This work reports the accuracy and capability of four data inversion methods for analyzing the data obtained by the particle size magnifier (PSM). The authors discovered that among all the data inversion methods, the expectation-maximization (EM) algorithm provides the best agreement with the test aerosol size distributions both experimentally and computationally. However, all four data inversion methods generated false sub-3 nm particle concentrations when the aerosols were larger than 3 nm, because of the limited resolution of the PSM in controlling the saturator flow rate. Suggestions regarding the PSM operation and data inversion were also given based on the findings

of this work. This work will be a very useful publication regarding the interpretation of the PSM data because the PSM is receiving wide applications in new particle formation studies. The reviewer recommends the publication of this manuscript after a few points regarding the resolution of the PSM and the data inversion methods are properly clarified.

Specific comments:

1. Resolution of the PSM: The reviewer understands that the resolution of the PSM should be defined based on the saturator flow rate instead of the particle size, since the relationship between the saturator flow rate and particle size are dependent on the chemical composition and charging state of the sub-3 nm particles. But it may still be helpful to translate the resolution in terms of particle size since this study does not consider the influence of the chemical composition and charging state. It was mentioned in the manuscript several times that the low resolution of the PSM led to the detection of false sub-3 nm particles even when the particles are above 3 nm. But one wonders how this "low resolution" (e.g. $\sim$ 1.0 in Page 11 Line 4) may relate to the resolution in terms of particle size. For example, in this study, does a resolution of $\sim$ 1.0 at 3.93 nm (based on the saturator flow rate) simply mean that particles of 3.93 nm can be detected by the PSM in the size bins between 3.93-1.97nm and 3.93+1.97 nm?

2. Stability of the non-negative least squares method: In this study, both the kernel function method and the H&A method used the non-negative least-squares method (probably the "lsqnonneg" function in MATLAB) to directly solve the particle size distributions. This function indeed can cause instabilities when the inversion matrix becomes complex. The reviewer wonders whether the authors could use the Twomey inversion algorithm to further refine the solution by using the results of the non-negative least-squares method as an initial guess. One can refer to Eqs. 3 and 4 of Markowski (1987) for further detailed calculation methods. In this way, the instability of the non-negative least squares method can be reduced. The smoothing algorithm could be disabled (neglecting Eqs. 6 and 7 of Markowski (1987)) if the authors are concerned

with its adverse influence on data inversion. Since the Twomey inversion method involves the iteration of linear equations, the computational expense should be low as well.

Reference: Markowski, G. R. (1987). "Improving Twomey's algorithm for inversion of aerosol measurement data." Aerosol Sci. Technol. 7(2): 127-141.

3. The detection of false sub-3 nm particles: The reviewer is quite puzzled by the detection of false sub-3 nm particles when the test aerosols were above 3 nm. Take the stepwise method for an example, theoretically, according to Eq. 4, $n_m$ would become 0 when the test aerosols were all above 3 nm. Even if we consider the limited resolution of the PSM for particles above 3 nm, the PSM should not report the detection of aerosols below 1.5 nm, which is shown in Figures 5 and 6. The reviewer wonders if the false detection of the sub-3 nm particles is related to the "error/uncertainty" in both the simulation and experiments, rather than the low resolution of the PSM. In addition, could the authors show the PSM-measured particle concentrations as a function of the saturator flow in Figure 6, similar to the one in Figure 5b, so that the error/uncertainty during the experiment could be evaluated?

Technical comments:

1. Page 3 Line 19: "... a regularization parameter... and the agreement with the PSM recorded data, ..." Was PSM data analyzed by using the Tikhonov regularization method?

2. Page 5 Line 17: "size ability" –> "sizing ability"?

3. Page 6 Line 3: "step-wising" –> "stepwise", same applies to the rest of the manuscript.

4. Eq. (4): Please check the unit of $n_m$. R should have a unit of $cm^{-3}$, and the denominator is dimensionless. Also, regarding the efficiency terms in the denominator, should they be eta(s_i,d_p,max) and eta(s_i+1,d_p,max), because the calculation is

specific for the ith and (i+1)th saturator flow rates?

5. Please check equations (6) and (8)-(10), they have some format issues on my computer.

6. Eq. (10): According to the definition of the H&A method, should the matrix Q have a dimension of J»I? Otherwise, please include some important steps converting the matrix into a square matrix.

7. Page 9 Line 23: "No./cm3" –> "cmˆ(-3)"?

8. Page 9 Line 30: What was the approximate time needed for the measurement in the stepping mode and how stable was the wire generator?

9. Page 15 Line 13: "sable" –> "stable"?

---

## Referee Comment (RC2) · Anonymous Referee #2 · 10 May 2018

The authors present four different data inversion methods for analyzing the experimental data obtained by the particle size magnifier (PSM). According to this study, the expectation-maximization (EM) algorithm shows the best agreement for the tested generated and simulated aerosol size distribution. Recommendations for reducing the measurement uncertainties like software/hardware improvements or change the scanning scheme of the saturator flow rate was suggested. The manuscript addresses an important topic and presents interesting results which especially can help for a better understanding of atmospheric new particle formation. The reviewer recommends the publication of this manuscript after some minor and technical corrections which are listed below.

[Figure]

Minor Comments:

The Reviewer is wondering how the different studied inversion algorithm will perform when the particle number concentration of the observed aerosol is strongly decreased like it is under atmospheric conditions? Have any considerations been made concerning the application onto atmospheric conditions?

Technical Corrections:

p.1, l.14 using diethylene glycol as "the" working fluid – remove "the"

p.6, l.22 eq. 6 - the uppercase subscriptions are not fully visible

p.7, l.4 eq. 8 J ? I – please check equation

p.7, l.10 eq. 9 - the uppercase subscriptions are not fully visible

p.7, l.11 the uppercase subscriptions are not fully visible

p.7, l.16 eq. 10 - the uppercase subscriptions are not fully visible

p.15, l.13 sable – stable

p.23, l.1 Figure 4 is full of information – Consider to increase the figure size or add an table with the stated concentrations.
* * *

---

## Author Comment (AC1) · 20 Jun 2018

Please see the supplement

Please also note the supplement to this comment:
https://www.atmos-meas-tech-discuss.net/amt-2018-35/amt-2018-35-AC1-supplement.zip

———————————————————

---

## Author Comment (AC2) · 20 Jun 2018

Please see the supplement

Please also note the supplement to this comment:
https://www.atmos-meas-tech-discuss.net/amt-2018-35/amt-2018-35-AC2-supplement.zip

---

## Author Response (AR1)

**Responses to Reviewers' Comments on Manuscript amt-2018-35**

**(Data inversion methods to determine sub-3 nm aerosol size distributions using the Particle Size Magnifier)**

We thank the reviewers for their help in improving this manuscript. We have addressed the comments in the following paragraphs and made corresponding changes in the revised manuscript. Comments are shown as *blue italic text* followed by our responses. Changes are highlighted in the revised manuscript and shown as underlined text in the responses. Line numbers and equation numbers quoted in the following responses correspond to those in the revised manuscript.

**Reviewer #1:**

1) Resolution of the PSM: The reviewer understands that the resolution of the PSM should be defined based on the saturator flow rate instead of the particle size, since the relationship between the saturator flow rate and particle size are dependent on the chemical composition and charging state of the sub-3 nm particles. But it may still be helpful to translate the resolution in terms of particle size since this study does not consider the influence of the chemical composition and charging state. It was mentioned in the manuscript several times that the low resolution of the PSM led to the detection of false sub-3 nm particles even when the particles are above 3 nm. But one wonders how this "low resolution" (e.g. ~ 1.0 in Page 11 Line 4) may relate to the resolution in terms of particle size. For example, in this study, does a resolution of ~ 1.0 at 3.93 nm (based on the saturator flow rate) simply mean that particles of 3.93 nm can be detected by the PSM in the size bins between 3.93-1.97nm and 3.93+1.97 nm?

Response: The resolution based on the particle size was not reported in the manuscript due to the lacking of an unambiguous definition based on the particle size. For a given particle size, the kernel function is determined by the saturator flow rate. Thus, the straightforward definition for sizing resolution is based on the saturator flow rate. To obtain a sizing resolution based on the particle size, one need to assume/define a relationship between the particle size and the saturator flow rate, e.g., relating the particle diameter to its corresponding saturator flow rate at the kernel function peak (the dashed line in Fig. 2). However, such a definition does not sufficiently indicate the size-resolving ability of the PSM because of the differences in the kernel peak heights and the asymmetric peak shape. To illustrate the size-resolving ability, here we propose a non-standard definition of particle size sizing resolution based on the particle size:

$$\operatorname{Res}_{dp} = d_p / (d_u - d_l) ,$$

 $\operatorname{Res}_{dp}$  is the sizng resolution based on the particle size.  $d_p$  is the particle diameter.  $d_u$  and  $d_l$  are determined according the following criterion: The saturator flow rates corresponding to the 90%

maximum detection efficiency for  $d_u$  and the 10% maximum detection efficiency for  $d_l$  are equal to the saturator flow rate corresponding to the 10% and 90% maximum detection efficiencies for  $d_p$ , respectively. It should be clarified that this nonstandard definition reports a lower resolution than the "standard" definition defined using the full width at half maximum. However, it can be approximately regarded that  $d_p$  can be detected by the PSM in the size bins between  $d_l$  and  $d_u$ . The relationship between  $\text{Res}_{dp}$  and  $d_p$  is shown in Fig. R1.

Fig. R1. The sizing resolution based on the particle size as a function of the particle size However, we prefer not to use this non-standard definition for the sizing resolution to avoid any potential confusion. Additionally, the possible size range reported by the PSM estimate using this resolution based on the particle size alone may not be accurate because the shape of the kernel functions are asymmetric.

We added "The 3.93 nm particles contribute to the signal for 2.17 nm particles when using the stepwise method (inferred from Fig. 1 and Fig. 2)." in line 11 Page 11 and "However, the resolution alone is not sufficient to indicate the possible reported size range when the PSM is measuring monodisperse particles because the kernel functions are asymmetric and the inversion method also affect the reconstructed peaks." in line 27 Page 5.

2. Stability of the non-negative least squares method: In this study, both the kernel function method and the H&A method used the non-negative least-squares method (probably the "lsqnonneg" function in MATLAB) to directly solve the particle size distributions. This function indeed can cause instabilities when the inversion matrix becomes complex. The reviewer wonders whether the authors could use the Twomey inversion algorithm to further refine the solution by using the results of the non-negative least-squares method as an initial guess. One can refer to Eqs. 3 and 4 of Markowski (1987) for further detailed calculation methods. In this way, the instability of the nonnegative least squares method can be reduced. The smoothing algorithm could be disabled (neglecting Eqs. 6 and 7 of Markowski (1987)) if the authors are concerned with its adverse influence on data inversion. Since the Twomey inversion method involves the iteration of linear equations, the computational expense should be low as well.

**Reference: Markowski, G. R. (1987). "Improving Twomey's algorithm for inversion of aerosol measurement data." Aerosol Sci. Technol. 7(2): 127-141.**

Response: We had tried both the Twomey's algorithm and the Twomey-Markowski algorithm using the result of the H&A method as the initial guess. Sometimes the Twomey's routine could not get a convergent result. Note that the iteration step in Twomey's algorithm (Eq. 7 in Twomey 1975) does not guarantee convergence. A relaxation factor can be applied to reduce the changes in each step and to increase the probability to obtain a convergent result. However, convergence is still not mathematically guaranteed even using a large relaxation factor. The Twomey's routine is exited when the normalized chi-square statistic is smaller than one, and an error tolerance factor was used to control the chi-square statistic (Eq. 8 in Markowski 1987). The convergence and the iteration result are affected by the error tolerance factor. For instance, using the error tolerance factor suggested in Buckley and Hogan (2017), i.e., 0.03 - 0.06, could not obtain a convergence result for some tests in this study.

One can get convergent results using the Twomey's algorithm via tuning the relaxation factor and the error tolerance factor. However, the aim of this study is to figure out a penitential inversion method to recover various aerosol size distributions measured in long-term observation. The Twmney's algorithm may not be one the best choices because the inverted result is affected by the factors and the convergence is not mathematically guaranteed.

3. The detection of false sub-3 nm particles: The reviewer is quite puzzled by the detection of false sub-3 nm particles when the test aerosols were above 3 nm. Take the stepwise method for an example, theoretically, according to Eq. 4, n\_m would become 0 when the test aerosols were all above 3 nm. Even if we consider the limited resolution of the PSM for particles above 3 nm, the PSM should not report the detection of aerosols below 1.5 nm, which is shown in Figures 5 and 6. The reviewer wonders if the false detection of the sub-3 nm particles is related to the "error/uncertainty" in both the simulation and experiments, rather than the low resolution of the PSM. In addition, could the authors show the PSM-measured particle concentrations as a function of the saturator flow in Figure 6, similar to the one in Figure 5b, so that the error/uncertainty during the experiment could be evaluated?

Response: The simulated/experimental error contributes to the reported false sub-3 nm particle concentrations. When assuming there is no error in the simulated particle concentration detected by the PSM, the EM algorithm, the H&A method, and the kernel function method report nearly zero sub-3 nm particle concentrations and the stepwise reports zero sub-1.5 nm particle concentration (indicated by the dash-dot lines in Fig. 5). The idea of Fig. 5 is to simulate the performance of the four inversion methods under the influence of experimental errors. The false particle size distribution reported by the stepwise method (indicated by the dash-dot lines in Fig. 5f) when assuming no error was because of neglecting the sizing resolution.

To clarified this, we revised the discussions in lines 6-14, page 12 as: "The simulated uncertainty is the main cause of the false sub-3 nm particle concentrations reported by the H&A method and the kernel function method in Fig. 5. When assuming that there is no error in the particle concentration detected by the PSM, the H&A method and the kernel function method report nearly no particles in the sub-3 nm size range. Different from the H&A method and the kernel function method that reported false results due to their instability, the step-wising method reported false particle size distributions when assuming there are no uncertainties (Fig. 5f). This is because the step-wising method assumes a simple one-to-one relationship between the saturator flow rate and the recovered particle diameter instead of accounting for the wide kernel function peaks. For sub-1.5 nm particles, the nonzero mean particle concentration reported by the stepwise method is due to the simulated uncertainties."

The particle concentrations detected by the PSM were shown in Fig. 6b. The discussions on the false particle concentrations in Fig. 6 was revised as "Based on both the simulating and experimental results, we conclude that the PSM may report false sub-3 nm particle size distributions when there are actually no sub-3 nm particles because of the uncertainties and the non-ideal data inversion methods, especially the step-wising method" in line 20, page 12.

**Technical comments:**

1. Page 3 Line 19: "...a regularization parameter ...and the agreement with the PSM recorded data, ..." Was PSM data analyzed by using the Tikhonov regularization method?

Response: We revised "PSM recorded data" as "recorded signals" in line 19, page 3.

2. Page 5 Line 17: "size ability" -> "sizing ability"?

Response: Done.

3. Page 6 Line 3: "step-wising" -> "stepwise", same applies to the rest of the manuscript.

Response: Done.

4. Eq. (4): Please check the unit of n\_m. R should have a unit of  $cm^{(-3)}$ , and the denominator is dimensionless. Also, regarding the efficiency terms in the denominator, should they be  $eta(s_i,d_p,max)$  and  $eta(s_i+1,d_p,max)$ , because the calculation is specific for the ith and (i+1)th saturator flow rates?

Response: Thanks. We revised Eq. (4) as

$$n_{m} = \frac{2(R_{i+1} - R_{i})}{\eta(s_{\max}, d_{i}) + \eta(s_{\max}, d_{i+1})} \times \frac{1}{d_{i} - d_{i+1}}$$

5. Please check equations (6) and (8)-(10), they have some format issues on my computer.

Response: We replaced those symbols that may lead to format issues.

**6. Eq. (10): According to the definition of the H&A method, should the matrix Q have a dimension of J»I? Otherwise, please include some important steps converting the matrix into a square matrix.**

Response: Eq. 10 is the key step to get the square matrix, **Q**. We added Eq. 9 and Eq. 11 and revised Eq. 12 to illustrate the main idea of the converting steps.

$$R_{i} = \sum_{j=1}^{J} \eta \left( s_{i}, d_{j} \right) \times n_{j} \times \Delta d_{j}, J \gg I$$
(2)

$$\mathbf{R}_{I\times I} = \mathbf{P}_{I\times J} \cdot \mathbf{n}_{J\times I} \tag{3}$$

Eq. 9 is the vector form for Eq. 8 and **P** is the matrix relating  $n_j$  and **R**. ...

$$n_{\rm j} \approx f\left(\mathbf{n}_{\rm i}, d_{\rm j}\right),\tag{4}$$

$$\mathbf{n}_{J\times I} \approx \mathbf{F}_{J\times I} \cdot \mathbf{n}_{I\times I} \,, \tag{5}$$

where *f* is the function relating  $n_j$  and  $\mathbf{n}_i$  ( $\mathbf{n}_i$  is a vector);  $n_j$  is the particle size distribution function at  $\underline{d}_i$ ;  $n_j$  is estimated using more than one single  $n_i$ ; and Eq. 11 is the vector form for Eq. 10. ...

$$\mathbf{R}_{I \times I} \approx \mathbf{P}_{I \times J} \cdot \mathbf{F}_{J \times I} \cdot \mathbf{n}_{I \times I} = \mathbf{Q}_{I \times I} \cdot \mathbf{n}_{I \times I}$$
(6)

**P** and **F** are determined according to Eq. 8-11 and thus **Q** is determined by  $\eta$ , *f*, and  $\Delta d_j$ ...

7. Page 9 Line 23: "No./cm3" -> "cm^(-3)"?

Response: We replaced all the No./ $cm^3$  with  $cm^{-3}$ .

8. Page 9 Line 30: What was the approximate time needed for the measurement in the stepping mode and how stable was the wire generator?

Response: We added "It took approximately 30 min to measure a particle size distribution" and "The relative standard deviation of the peak particle concentration measured by the DMA-FCE system was within ~10%" in lines 4-7, page 10.

9. Page 15 Line 13: "sable" -> "stable"?

Response: Done.

**Reviewer #2:**

The Reviewer is wondering how the different studied inversion algorithm will perform when the particle number concentration of the observed aerosol is strongly decreased like it is under atmospheric conditions? Have any considerations been made concerning the application onto atmospheric conditions?

Response: Theoretically, none of the studied inversion methods will report sub-3 nm particles when the measured particle concentration decreases monotonically with the increasing saturator flow rate. When measuring particles around 7.3 nm, a gradually decreasing particle concentration was observed due to the instability of the wire generator. The sub-3 nm particle concentrations inverted using the studied methods were zero/negligible.

The uncertainties/errors of the observed raw particle concentration in atmosphere are usually larger than in the laboratory. Thus, it is more complicated to test the performance of the inversion methods under various atmospheric conditions. An ongoing study is focusing on the comparisons of the inversion methods using atmospheric observation data and the results obtained using different instruments.

*Technical Corrections: p.1, l.14 using diethylene glycol as "the" working fluid – remove "the" p.15, l.13 sable – stable*

Response: Done.

p.6, l.22 eq. 6 - the uppercase subscriptions are not fully visible p.7, l.4 eq. 8 J? I – please check equation p.7, l.10 eq. 9 - the uppercase subscriptions are not fully visible p.7, l.11 the uppercase subscriptions are not fully visible p.7, l.16 eq. 10 - the uppercase subscriptions are not fully visible

Response: We replaced those symbols that may lead to format issues.

*p.23, l.1 Figure 4 is full of information – Consider to increase the figure size or add a table with the stated concentrations.*

Response: We moved the particle concentrations to Table 1 and removed the legends in Fig. 4(b),

4(c), and 4(d).

[revised manuscript text omitted]